# Synergistic Strategies for KMT2A-Rearranged Leukemias: Beyond Menin Inhibitor

**DOI:** 10.3390/cancers16234017

**Published:** 2024-11-29

**Authors:** Sandra Cantilena, Mohamed AlAmeri, Noelia Che, Owen Williams, Jasper de Boer

**Affiliations:** 1Hemispherian AS, 0585 Oslo, Norway; sandra.cantilena@hemispherian.com; 2Cancer Section, Development Biology and Cancer Programme, UCL GOS Institute of Child Health, London WC1N 1EH, UK; 3Department of Health—Abu Dhabi, Abu Dhabi 20224, United Arab Emirates; dalameri@doh.gov.ae; 4Australian & New Zealand Children’s Haematology/Oncology Group, Melbourne, VIC 3052, Australia; 5Australia & Hudson Institute of Medical Research, Melbourne, VIC 3168, Australia

**Keywords:** KMT2A-rearranged leukemia, acute myeloid leukemia, menin inhibitors, therapeutic resistance, targeted therapies, combination therapies

## Abstract

KMT2A-rearranged leukemias are an aggressive form of leukemia with high relapse rates, even after intensive treatment. While menin inhibitors have shown promise in early clinical trials, many patients eventually relapse due to resistance. Here, we explore alternative strategies to be used in combination with menin inhibitors to improve patient outcomes. These include targeting proteins and pathways essential for leukemia progression. The goal is to enhance the effectiveness of treatments, reduce relapse rates, and provide more durable responses for patients.

## 1. Introduction

Acute myeloid leukemia (AML) is a hematologic malignancy characterized by the clonal expansion of undifferentiated hematopoietic progenitor cells, leading to bone marrow failure and a rapidly progressive clinical course. Among the diverse genetic alterations associated with AML, *KMT2A* rearrangements (*KMT2Ar*) represent a particularly aggressive and therapeutically challenging subset [1,2,3]. These rearrangements, resulting from chromosomal translocations involving the *KMT2A* (MLL) gene at 11q23, occur in approximately 5–10% of all AML cases and are most commonly observed in pediatric patients [4]. *KMT2A*r leukemias are associated with a poor prognosis, marked by high relapse rates and dismal overall survival despite intensive chemotherapy and hematopoietic stem cell transplantation [5,6,7].

Current therapeutic options for *KMT2A*r AML are limited and have not significantly improved outcomes for these patients over the past few decades [8]. The heterogeneity of *KMT2A*r leukemias contributes to their resistance to conventional therapies [9], underscoring the urgent need for novel, targeted therapeutic approaches. In recent years, advances in our understanding of the epigenetic and transcriptional dysregulation driven by KMT2A-fusion proteins, as summarized in Figure 1, have led to the development of several targeted therapies, including menin inhibitors, which specifically disrupt the pathogenic interaction between menin and KMT2A-fusion proteins [10,11,12,13]. While AMLs with NPM1 mutations are generally associated with a favorable prognosis in the absence of FLT3 mutations, their prognosis becomes intermediate when accompanied by FLT3-ITD mutations. Moreover, the prognosis of relapse and/or refractory NPM1-mutated acute myeloid leukemias is not very encouraging, with median overall survival around 5–6 months [14,15]. Menin inhibitors have shown promising preclinical activity and early clinical efficacy in patients with KMT2Ar AML and NPM1-mutated AML. However, the emergence of resistance to menin inhibitors underscores the critical need for combination therapies that target multiple oncogenic pathways to achieve more durable remissions [11,12]. This paper explores the potential of integrating menin inhibitors with other targeted therapies, such as inhibitors of DOT1L, BRD4, KMT2A-fusion proteins, MYC, and c-MYB. By dissecting the mechanistic roles of these targets and the evidence supporting their combination with menin inhibitors, we aim to provide a comprehensive overview of the emerging therapeutic landscape for *KMT2A*r AML and propose novel strategies to enhance treatment efficacy and overcome resistance.

## 2. Mechanism of Action and Efficacy of Menin Inhibitors

AML is a complex and heterogeneous disease characterized by the clonal expansion of hematopoietic progenitor cells. Despite advances in molecular diagnostics and targeted therapies, AML remains difficult to treat due to various cytogenetic and epigenetic alterations. Recent therapeutic developments have focused on mutation-specific agents, such as FLT3 inhibitors [16], IDH inhibitors [17], as well as broader-targeted agents, such as the BCL-2 inhibitor venetoclax [18]. However, AML subsets driven by KMT2A rearrangements and NPM1 mutations (NPM1mt) continue to present significant treatment challenges.

Menin inhibitors have emerged as a promising therapeutic approach for these specific AML subsets. Menin, a protein encoded by the *MEN1* gene, plays a critical role in the leukemogenic process associated with *KMT2Ar* and *NPM1mt*. The pathogenic role of menin involves its interaction with the KMT2A complex, which binds to chromatin and regulates the expression of oncogenes such as *HOXA9* and *MEIS1*. Additionally, menin acts as a molecular adaptor, linking KMT2A proteins with LEDGF (lens epithelium-derived growth factor) on cancer-associated target genes, forming the menin–KMT2A–LEDGF complex, which is essential for KMT2A-dependent transcription and leukemic transformation [19]. Overexpression of these genes in *KMT2Ar* and *NPM1mt* leukemias promotes a differentiation block and maintains a stem-cell-like state in leukemic cells.

Menin inhibitors disrupt the interaction between menin and KMT2A, which is required for the binding of KMT2A-fusion proteins to their target genes. In *NPM1mt* leukemia, the interaction between menin and KMT2A is crucial for the binding of mutant NPM1c to chromatin. This cooperation between NPM1c and KMT2A sustains the transcription of key oncogenes like *MEIS1* and *HOX* cluster genes, which are vital for maintaining the stem-cell-like state of leukemia cells [20,21]. By inhibiting the menin–KMT2A interaction, menin inhibitors reduce the chromatin binding and oncogenic effects of NPM1c, leading to decreased transcription of leukemia-promoting genes, ultimately promoting differentiation and reducing leukemic cell proliferation [20,21]. This disruption results in the loss of target gene binding and inhibits the transcriptional programs driven by these oncogenes. This disruption leads to the differentiation of leukemic cells and apoptosis without significantly affecting normal hematopoiesis. Studies have supported the therapeutic potential of menin inhibitors, demonstrating their efficacy in reversing aberrant gene expression and inducing leukemic regression in mouse models of *KMT2Ar* and *NPM1mt* AML.

## 3. Clinical Development of Menin Inhibitors

Several small molecule menin inhibitors have been developed, showing promising early results in clinical trials:**Revumenib (SNDX-5613):** The AUGMENT-101 trial (NCT04065399) evaluated revumenib in a phase I/II study involving patients with relapsed/refractory (R/R) acute leukemia harboring *KMT2A* rearrangements or *NPM1* mutations. The interim analysis demonstrated clinical activity, with 24% of *KMT2A*r AML patients achieving complete remission (CR) or CR with partial hematologic recovery (CRh) [22]. Menin mutations as a resistance mechanism were also noted, emphasizing the need for continued monitoring and combination strategies [22,23].**Ziftomenib (KO-539):** The KOMET-001 trial (NCT04067336) investigated ziftomenib in patients with R/R AML. Early reports indicated a higher response rate in *NPM1*-mutated AML compared with *KMT2A*r AML, with CR rates of 35% and 5.6%, respectively [24,25]. This differential response suggests a possible shift in the trial’s focus toward *NPM1*-mutated AML patients in subsequent phases.**DSP-5336:** The ongoing phase I/II trial (NCT04988555) explores DSP-5336 in adult patients with R/R AML and ALL, specifically targeting those with *KMT2A*r and *NPM1* mutations. Early data indicate significant variance in efficacy, with the CR and CRh rate reaching 44% in the *NPM1*mt group, compared to only 8% in the *KMT2A*r cohort [26]. These findings highlight a potentially greater benefit of DSP-5336 in patients with *NPM1* mutations, guiding further investigation and clinical application in this subgroup.**Bleximenib:** The ongoing phase I/II trial (NCT04811560) is evaluating bleximenib (formerly known as JNJ-75276617), a menin–KMT2A inhibitor, in patients with relapsed/refractory (R/R) acute leukemia with *KMT2A* or *NPM1* alterations. As of April 2023, 58 patients were treated, with an overall response rate (ORR) of 50% at the highest dose level (90 mg BID), including complete remissions [27]. Common treatment-related adverse events included differentiation syndrome and cytopenias. Preliminary biomarker data showed reductions in menin–KMT2A target genes, including MEIS1, HOXA9, and FLT3, as well as the induction of genes associated with differentiation, such as ITGAM and MNDA. Dose escalation continues to determine the recommended phase 2 dose (RP2D).**DS-1594:** In the phase I/II trial (NCT04752163), DS-1594 is being studied both as a monotherapy and in combination with azacytidine and venetoclax or mini-HCVD (cyclophosphamide, vincristine, dexamethasone). Preclinical data suggested efficacy, but clinical results are awaited.**BMF-219:** The COVALENT-101 trial (NCT05153330) is a phase I study investigating BMF-219, a covalent menin inhibitor, in patients with AML, ALL, multiple myeloma, diffuse large B-cell lymphoma, and chronic lymphocytic leukemia. Initial findings reported complete remission in two out of five AML patients, with no dose-limiting toxicities observed.

Preliminary analyses suggest that NPM1-mutated AML responds better to menin inhibitors compared with KMT2Ar leukemias, indicating potential alternative or multiple resistance mechanisms in these subtypes. Understanding these differences is crucial for developing effective combination therapies and optimizing treatment strategies to overcome resistance and improve outcomes for patients with these challenging forms of AML.

### Combination Trials of Menin Inhibitors in AML

Recent advancements in menin inhibition have shifted the focus towards combination therapies to enhance therapeutic efficacy and mitigate resistance. These combinations primarily fall into two categories: regimens combining menin inhibitors with conventional chemotherapy and those leveraging venetoclax-based apoptotic pathways. Notably, the synergy observed in these combinations reflects complementary mechanisms of action targeting distinct oncogenic pathways in AML.

Chemotherapy-based trials have demonstrated promising results when paired with menin inhibitors. For example, the KOMET-007 trial, which includes multiple arms, explored the addition of ziftomenib to the standard 7+3 induction regimen (cytarabine and daunorubicin) for newly diagnosed (ND) AML with NPM1 mutations or KMT2A rearrangements. This arm reported complete remission composite (CRc) rates of 93% in NPM1-mutated and 78% in KMT2Ar AML, with high minimal residual disease (MRD) negativity rates of up to 100% in responders [28]. Similarly, the Phase 1b study of bleximenib combined with the 7+3 regimen targeted ND AML, achieving an overall response rate (ORR) of 93% and CR/CRh in 79% of patients, with NPM1-mutated cases showing slightly better outcomes than KMT2Ar cases [29]. Pediatric trials such as APAL2020K (ITCC-101) have paired ziftomenib with the FLA regimen (fludarabine and cytarabine) for relapsed/refractory (R/R) leukemias, focusing on safety, pharmacokinetics, and preliminary efficacy data, with detailed response outcomes still pending [30].

Venetoclax-based combinations have also gained significant attention. The SAVE trial evaluated an all-oral regimen of revumenib combined with venetoclax and decitabine/cedazuridine (ASTX727) for R/R AML with high-risk mutations, achieving an ORR of 88%, with 58% achieving CR/CRh and MRD negativity in 93% of CR/CRh responders [31]. Additionally, the KOMET-007 trial included an arm investigating ziftomenib with venetoclax and azacitidine in R/R AML patients. For menin inhibitor-naive cases, NPM1-mutated AML achieved CRc rates of 80% at a 200 mg dose and 50% at 400 mg, while KMT2Ar AML showed lower response rates of 29% and 17% at respective doses [32]. In another Phase 1 study, tuspetinib paired with venetoclax demonstrated a CRc rate of 30% in venetoclax-naive R/R AML patients, with promising results in FLT3-mutated AML subtypes (CRc of 37.5%) [33].

These combination strategies underscore the growing recognition that single-agent menin inhibition may be insufficient to achieve durable remissions, particularly in genetically heterogeneous AML subsets. By integrating chemotherapy and/or BCL-2 inhibition, these regimens aim to dismantle leukemia survival networks more comprehensively. However, resistance to menin inhibitors remains a significant clinical challenge, necessitating further investigation into the underlying mechanisms and novel strategies to address them.

## 4. Resistance Mechanisms to Menin Inhibitors

Menin inhibitors have shown efficacy in treating AML with KMT2Ar and NPM1mt. However, resistance remains a significant hurdle, constraining clinical effectiveness. Understanding both genetic and epigenetic resistance mechanisms is crucial for developing strategies that enhance therapeutic durability and patient outcomes.

### 4.1. Genetic Resistance Mechanisms

Approximately 40% of cases of resistance to menin inhibitors have been attributed to specific mutations in the MEN1 gene, particularly at residues M327, G331, or T349. These mutations decrease the binding affinity of menin inhibitors by altering critical binding sites, thereby preventing effective drug–target interactions. Structural analyses reveal that these residues are positioned near W346, a key amino acid involved in stabilizing inhibitor binding. Mutations, such as M327I, interfere with hydrogen bonds that anchor the inhibitor, causing steric clashes that reduce drug efficacy [23].

These mutations have been shown to evolve under selective pressure from menin inhibitors. Studies using patient-derived xenograft models and cellular assays identified additional resistance mutations, such as G331D and S160C, which similarly impair inhibitor binding without affecting KMT2A association with menin. This suggests that while the menin–KMT2A interaction remains intact, inhibitor binding is disrupted, allowing leukemic transcriptional programs to continue uninhibited [23].

### 4.2. Epigenetic and Alternative Resistance Mechanisms

Resistance to menin inhibitors in KMT2Ar leukemias often arises from epigenetic mechanisms and alternative cellular pathways rather than solely genetic mutations in the MEN1 gene. Studies show that inhibiting the menin–KMT2A interaction can lead leukemia cells to shift their transcriptional programs, compensating for the lost interaction [34]. The activation of alternative transcriptional programs via the MLL3/4-UTX complex [12,35] can trigger dynamic shifts in KMT2A oncoprotein binding, which may activate non-canonical lineage programs, such as transitioning from a lymphoid to a myeloid lineage, contributing to leukemia’s ability to resist therapy [36]. These complex epigenetic and transcriptional adaptations contribute to resistance, and recent research has uncovered an additional mechanism involving the histone acetyltransferase KAT6A [37].

In these resistant cells, key KMT2A target genes, such as MEIS1 and HOXA, remain suppressed, and although the menin–KMT2A complex is successfully displaced from chromatin by the inhibitors, leukemia persists. This suggests that displacing menin alone is not sufficient to prevent the epigenetic reprogramming driving the resistance. Notably, although the leukemia cells no longer express certain KMT2A-regulated genes, they still rely on the KMT2A-fusion protein for survival. Inactivation of KAT6A has been shown to reverse this resistance, suggesting that targeting both KAT6A and the menin–KMT2A interaction can restore the cells’ sensitivity to menin inhibitors and enhance the overall therapeutic response [37].

Polycomb repressive complexes, specifically PRC1.1 and PRC2.2, also play a role in menin inhibitor resistance [12,38]. Depletion of *PRC1.1* components such as PCGF1, BCOR, and RYBP enhances resistance, potentially due to the aberrant activation of *MYC*, a critical oncogene co-regulated by the menin–KMT2A and PRC1.1 complexes [39]. Loss of PRC1.1 diminishes the monocyte differentiation signature, making cells more susceptible to *BCL-2* inhibitors, such as venetoclax [40].

Understanding these multifaceted resistance mechanisms is essential for developing combination therapies that target both the primary leukemic drivers and the pathways contributing to resistance, potentially improving treatment outcomes.

Resistance to menin inhibitors is driven by a combination of epigenetic reprogramming, chromatin remodeling, and activation of alternative oncogenic pathways rather than a single genetic event. Key contributors to this resistance include the polycomb repressive complexes PRC1.1 and PRC2.2, and the histone acetyltransferase *KAT6A*. Inactivation of these components, alongside targeting the menin–KMT2A interaction, could enhance therapeutic efficacy and restore sensitivity to treatment.

Menin inhibitors, such as revumenib, have shown considerable promise in treating AML characterized by *KMT2A*r and *NPM1*mt. However, resistance to these therapies remains a significant obstacle, impacting their overall clinical effectiveness and patient outcomes. Understanding the mechanisms of resistance is crucial for devising strategies to counteract these challenges and improve therapeutic results.

## 5. Alternative and Synergistic Therapeutic Strategies

While menin inhibitors have shown significant promise in targeting *KMT2A*r and *NPM1*mut AML, the complexity and redundancy of oncogenic pathways in these leukemias often lead to resistance, underscoring the need for combination therapies. To address this challenge, a variety of alternative therapeutic strategies have been explored, each targeting different aspects of the leukemic transcriptional and epigenetic machinery.

Importantly, cells that lost their dependency on menin remained addicted to the KMT2A-fusion protein, highlighting the central role of KMT2A-fusion protein in sustaining leukemic survival [37]. The rationale for combining these alternative therapies with menin inhibitors lies in their complementary mechanisms of action. By simultaneously targeting multiple oncogenic drivers, these combinations aim to dismantle the complex networks that support leukemia survival and proliferation, thereby enhancing therapeutic efficacy and overcoming resistance mechanisms. The following sections explore the roles of DOT1L, BRD4, chromatin remodeling complexes, KMT2A-fusion proteins, MYC, and c-MYB in leukemogenesis, strategies for their inhibition, and the potential synergistic effects when combined with menin inhibitors (see Figure 2). These insights form a compelling argument for pursuing further preclinical and clinical investigations into these combination therapies.

### 5.1. Targeting DOT1L: Role, Strategies of Inhibition, and Potential Synergy with Menin Inhibitors

#### 5.1.1. The Role of DOT1L in KMT2Ar Leukemias

DOT1L is a histone methyltransferase responsible for methylating histone H3 at lysine 79 (H3K79), a modification crucial for the transcriptional activation of genes involved in leukemogenesis [41]. In *KMT2A*r leukemias, DOT1L plays a vital role in sustaining the oncogenic transcriptional programs driven by KMT2A-fusion proteins. This methylation process is essential for maintaining the expression of key oncogenes, including *HOXA9* and *MEIS1*, which promote leukemic cell survival and proliferation.

#### 5.1.2. Strategies for Inhibiting DOT1L

Inhibition of DOT1L has emerged as a promising therapeutic strategy for KMT2Ar leukemias. DOT1L inhibitors, such as EPZ-5676 (pinometostat), specifically target the methyltransferase activity of DOT1L, leading to reduced H3K79 methylation and downregulation of KMT2A-fusion target genes [42]. Despite the promising mechanism, clinical trials of DOT1L inhibitors as monotherapy have shown limited efficacy, particularly in achieving durable remissions, likely due to the redundancy of oncogenic pathways in KMT2Ar leukemias [42].

#### 5.1.3. Potential Synergy with Menin Inhibitors

Preclinical studies have demonstrated that combining DOT1L inhibitors with menin inhibitors significantly enhances anti-leukemic effects in *KMT2A*r leukemias. For example, when combined with menin inhibitors, DOT1L inhibitors not only suppress the methylation of H3K79 but also downregulate the expression of the KMT2A target genes, leading to more profound leukemic cell death. This synergy is particularly evident in models of acute lymphoblastic leukemia (ALL), where combination treatment induces rapid and dose-dependent cell death. In acute myeloid leukemia (AML) models, the combination also promotes differentiation, a key process blocked in *KMT2A*r leukemias, thus overcoming resistance frequently observed with menin inhibition alone. These findings provide a strong rationale for pursuing clinical trials that combine DOT1L and menin inhibitors to enhance treatment efficacy and achieve more durable remissions [43,44].

### 5.2. Targeting BRD4: Role, Strategies of Inhibition, and Potential Synergy with Menin Inhibitors

#### 5.2.1. The Role of BRD4 in KMT2Ar Leukemias

BRD4, a member of the bromodomain and extraterminal (BET) protein family, is crucial for regulating gene expression by recognizing acetylated lysines on histones and recruiting transcriptional machinery to active enhancers and super-enhancers. In *KMT2A*r leukemias, BRD4 is often associated with super-enhancers that drive the expression of oncogenes such as *MYC*, *BCL2,* and *CDK6*, which are vital for leukemic cell survival and proliferation [45].

#### 5.2.2. Strategies for Inhibiting BRD4

BET inhibitors, including JQ1 and OTX015, target BRD4 by blocking its bromodomains, preventing interaction with acetylated histones. BRD4 acts as a scaffold protein that recruits transcription factors, acetylated histones, and the transcriptional machinery to cis-regulatory elements, thereby facilitating RNA polymerase II activation and transcription elongation [46,47]. This inhibition disrupts the transcriptional programs sustained by BRD4, including its role in maintaining the activity of super-enhancers associated with key oncogenes such as *BCL2, CDK6, CDK4, MYC,* and *CCND1* [48,49]. Consequently, BET inhibitors lead to reduced expression of these oncogenes, induce differentiation, and trigger apoptosis in leukemic cells. However, the efficacy of BET inhibitors as monotherapies is often limited due to the complex and redundant nature of oncogenic signaling in KMT2Ar leukemias.

#### 5.2.3. Potential Synergy with Menin Inhibitors

Recent studies have explored the synergistic potential of combining BRD4 inhibitors with menin inhibitors. In preclinical models, this combination has been shown to enhance the anti-leukemic effects by targeting both upstream regulation and downstream activation of oncogenic transcriptional networks. Specifically, menin inhibitors reduce the interaction with KMT2A, while BRD4 inhibitors further suppress transcriptional activation of key oncogenes [48,49]. This dual blockade effectively collapses the oncogenic transcriptional network, leading to more pronounced anti-leukemic effects and offering a compelling strategy to overcome resistance mechanisms that might arise from single-agent therapies. The identification of synergistic effects in these preclinical models provides a strong foundation for considering clinical trials that combine BET and menin inhibitors [45].

### 5.3. Direct Targeting of KMT2A-Fusion Proteins: Role, Strategies of Inhibition, and Potential Synergy with Menin Inhibitors

#### 5.3.1. The Role of KMT2A-Fusion Proteins in Leukemogenesis

KMT2A-fusion proteins, resulting from rearrangements of the *KMT2A* gene, are central to the pathogenesis of *KMT2A*r leukemias. These fusion proteins disrupt normal hematopoietic differentiation and promote the maintenance of leukemic stem cells by dysregulating gene expression programs critical for leukemic cell survival and proliferation.

#### 5.3.2. Strategies for Direct Inhibition of KMT2A-Fusion Proteins

Recent advances have identified disulfiram, traditionally used to treat alcohol dependence, as a potent inhibitor of KMT2A-fusion proteins. Disulfiram directly targets the N-terminal CXXC domain of KMT2A-fusion proteins, which is essential for their DNA-binding activity [50,51]. This inhibition leads to the degradation of KMT2A-fusion proteins and subsequent reduction in the transcriptional activity of oncogenes such as *HOXA9* and *MEIS1*, both of which are pivotal in sustaining the leukemic phenotype.

#### 5.3.3. Potential Synergy with Menin Inhibitors

Combining direct KMT2A-fusion protein inhibitors such as disulfiram with menin inhibitors offers a promising therapeutic strategy. While disulfiram induces the degradation of KMT2A-fusion proteins, menin inhibitors block the KMT2A–menin interaction necessary for the fusion proteins’ oncogenic activity. This dual approach not only enhances the depletion of KMT2A-fusion proteins but also prevents the re-establishment of oncogenic transcriptional programs, potentially reducing resistance development. An analogous successful approach can be seen in the treatment of acute promyelocytic leukemia (APL), where the combination of all-trans retinoic acid (ATRA) and arsenic trioxide (ATO) targets different moieties of the PML-RARα fusion oncoprotein [52,53]. ATRA induces terminal differentiation, while ATO promotes degradation of the fusion protein, leading to synergistic effects that overcome resistance and achieve deeper remissions [54]. By disrupting both the structural integrity and functional interactions of the fusion proteins, the combination of disulfiram and menin inhibitors could achieve similar deeper remissions in patients with KMT2Ar leukemias. These findings highlight the potential for rigorous preclinical studies to pave the way for clinical trials testing this combination.

### 5.4. MYC Inhibition: Role, Strategies of Inhibition, and Potential Synergy with Menin Inhibitors

#### 5.4.1. The Role of MYC in KMT2Ar Leukemias

MYC is a critical oncogene frequently upregulated in *KMT2A*r leukemias, where it plays a key role in maintaining the proliferation and survival of leukemic cells. MYC drives oncogenesis by promoting the expression of genes involved in cell growth, metabolism, and survival, making it an essential target for therapeutic intervention in these leukemias.

#### 5.4.2. Strategies for Direct Inhibition MYC

Direct inhibition of MYC has been challenging due to its “undruggable” nature, but several strategies have emerged to inhibit its function indirectly. BET inhibitors, such as JQ1, disrupt MYC’s interaction with binding partners and reduce MYC levels, impairing leukemic cell growth [49]. Other approaches include disrupting MYC–MAX interaction [55] and novel compounds such as Tosyl Chloride-Berbamine (TCB), which eliminates MYC-positive leukemia by targeting CaMKIIγ [56]. Additionally, OmoMYC or OMO-103, a peptide that blocks MYC binding to promoters, has shown anticancer activity with minimal side effects [57]. APTO-253, currently in phase I trials, also indirectly reduces MYC expression in AML and myelodysplastic syndrome [56].

#### 5.4.3. Potential Synergy with Menin Inhibitors

Combining MYC inhibitors with menin inhibitors offers a powerful therapeutic strategy. Menin inhibitors reduce MYC expression by disrupting the KMT2A–menin interaction, while MYC inhibitors can block any residual MYC activity that persists despite menin inhibition. This dual targeting approach could dismantle the MYC-driven oncogenic network at multiple levels, leading to a more profound suppression of leukemic cell survival and proliferation. Moreover, this combination could overcome resistance mechanisms that might arise from targeting either pathway alone. Preclinical studies that explore the efficacy of this combination could provide critical insights into its potential for clinical application.

### 5.5. Targeting c-MYB: Role, Strategies of Inhibition, and Potential Synergy with Menin Inhibitors

#### 5.5.1. The Role of c-MYB in KMT2Ar Leukemias

c-MYB is a transcription factor that plays a crucial role in regulating hematopoiesis, particularly in maintaining hematopoietic stem cells and controlling myeloid differentiation. In *KMT2A*r leukemias, c-MYB is essential for sustaining the leukemic phenotype by driving transcriptional programs that support the self-renewal and proliferation of leukemic cells.

#### 5.5.2. Strategies for Inhibiting c-MYB

Direct inhibition of c-MYB has been difficult due to the nature of transcription factors, but recent advances have identified promising strategies, including small molecule inhibitors that induce c-MYB degradation. Mebendazole, for example, has been shown to promote proteasomal degradation of c-MYB, disrupting its oncogenic transcriptional programs and reducing leukemic cell viability [58]. Additionally, disrupting the interaction between MYB and its co-activators, such as CBP/P300, has been found to remodel oncogenic MYB complexes, inducing differentiation and apoptosis [59,60]. The MYB-C/EBPβ-p300 module also represents a potential target for anti-tumor drugs, with natural products showing inhibitory effects [59,60].

#### 5.5.3. Potential Synergy with Menin Inhibitors

Given the central role of c-MYB in *KMT2A*r leukemias, combining c-MYB inhibitors with menin inhibitors could offer a powerful therapeutic approach. Menin inhibitors reduce MYC levels by disrupting the KMT2A–menin interaction, while c-MYB inhibitors directly impair the transcriptional machinery that drives leukemic cell survival. This combination could effectively target multiple key oncogenic pathways, providing a robust and comprehensive treatment strategy that could overcome resistance mechanisms and lead to more sustained remissions. Future preclinical models should evaluate this combination strategy, with successful studies potentially leading to clinical trials.

### 5.6. Targeting Chromatin Remodeling: Role, Strategies of Inhibition, and Potential Synergy with Menin Inhibitors

#### 5.6.1. The Role of Chromatin Remodeling Complexes in Leukemogenesis

Chromatin remodeling complexes, including the BAF (BRG1/BRM-associated factor) complex and the histone demethylase KDM4C, are key regulators of gene expression in KMT2Ar leukemias [61]. The BAF complex, driven by BRG1 (SMARCA4) or BRM (SMARCA2), enables transcription factors such as PU.1 and MYC to bind enhancers and promoters, sustaining oncogenic transcriptional programs essential for leukemia cell survival.

KDM4C, on the other hand, removes repressive H3K9me3 marks, facilitating the expression of critical leukemogenic genes such as HOXA9 and MYC. KMT2A fusion proteins exploit KDM4C to drive epigenetic reprogramming necessary for leukemic proliferation and maintenance [62]. Together, these complexes create a permissive chromatin environment that supports aberrant transcriptional networks, underscoring their therapeutic importance as dual targets in disrupting leukemia’s oncogenic machinery.

#### 5.6.2. Strategies for Inhibiting Chromatin Remodeling Complexes

Recent developments have identified FHD-286 as a selective inhibitor of BRG1/BRM ATPases. Preclinical studies demonstrated that FHD-286 induces differentiation and apoptosis in leukemia cells harboring KMT2Ar or NPM1mut. By disrupting BRG1/BRM-mediated chromatin accessibility, FHD-286 downregulates key oncogenic transcriptional networks, including those driven by MYC and PU.1, while enhancing the activity of differentiation-promoting genes [61].

Another promising approach involves targeting histone demethylases, such as KDM4C, which is recruited by KMT2A fusion proteins to modulate H3K9 methylation. Inhibitors such as SD70 have shown efficacy in reducing leukemic cell survival and downregulating MYC-driven transcriptional programs [62].

#### 5.6.3. Potential Synergy with Menin Inhibitors

Combining chromatin remodeling inhibitors with menin inhibitors represents a compelling therapeutic strategy. Preclinical studies revealed that the dual targeting of BRG1/BRM complexes and menin with FHD-286 and MI-503, respectively, leads to synergistic anti-leukemic effects. This combination enhances apoptosis and differentiation in leukemia models, effectively reducing disease burden and improving survival in xenograft models [61].

Similarly, the combination of SD70 with menin inhibitors has been shown to significantly downregulate MYC expression and disrupt its downstream transcriptional network. This dual inhibition strategy not only enhances apoptosis in leukemia cells but also spares normal hematopoietic stem cells, underscoring its therapeutic potential [62].

These findings highlight the potential of targeting chromatin remodeling complexes in combination with menin inhibitors to overcome resistance and achieve deeper remissions in KMT2Ar leukemias. Further preclinical and clinical studies are warranted to optimize these combination strategies and validate their efficacy and safety in diverse patient populations.

## 6. Discussion

The treatment of *KMT2A*r acute myeloid leukemia (AML) remains one of the most formidable challenges in hematologic oncology, primarily due to the aggressive nature of these leukemias and their tendency to resist conventional therapies. Menin inhibitors have emerged as a promising therapeutic strategy by specifically targeting the critical menin–KMT2A interaction that drives the oncogenic transcriptional programs in these leukemias. However, the complexity of the disease suggests that menin inhibition alone may not suffice to achieve long-term remission, given the emergence of genetic and non-genetic resistance mechanisms. This necessitates a broader approach that leverages combination therapies targeting multiple oncogenic pathways concurrently.

The role of DOT1L in maintaining the transcriptional activity of KMT2A-fusion proteins provides a compelling case for its combination with menin inhibitors. DOT1L’s enzymatic function, which methylates histone H3 at lysine 79 (H3K79), sustains the expression of *HOXA9* and *MEIS1*—oncogenes. Menin inhibitors, by disrupting the menin–KMT2A interaction, reduce the transcriptional activation of these same oncogenes. The dual inhibition of both DOT1L and menin disrupts complementary aspects of the transcriptional machinery, potentially leading to more profound and sustained anti-leukemic effects. This combination targets both the chromatin landscape and the transcriptional network, thereby crippling the leukemia cells’ ability to maintain their malignant state. The preclinical success of this strategy underscores its potential, yet the real challenge lies in translating these findings into clinical practice, where the efficacy, safety, and optimal dosing of this combination must be rigorously evaluated.

BRD4, a key regulator of super-enhancers and a driver of MYC expression, presents another attractive target for combination with menin inhibitors. The inhibition of BRD4 with BET inhibitors disrupts the transcriptional machinery at the level of super-enhancers, which are crucial for maintaining the high expression of MYC and other oncogenes in *KMT2A*r leukemias. Menin inhibitors complement this by directly reducing MYC transcription through the disruption of the menin–KMT2A interaction. The synergy between BET inhibitors and menin inhibitors lies in their ability to attack the transcriptional architecture of leukemic cells at multiple levels—both upstream at the super-enhancer level and downstream at the promoter level. This multi-level disruption could be particularly effective in preventing leukemia from compensating through alternative transcriptional programs, thus reducing the likelihood of resistance. However, this approach also requires careful balancing to avoid excessive toxicity, given the central role of BRD4 in normal cellular processes.

The strategy of directly targeting KMT2A-fusion proteins themselves, as with Disulfiram, offers a more aggressive approach by aiming to degrade the very proteins that drive leukemogenesis. The combination of such direct inhibitors with menin inhibitors could prove highly effective, as it would simultaneously dismantle the structural integrity of the fusion proteins and inhibit their functional interactions necessary for oncogenesis. This approach not only enhances the depletion of KMT2A-fusion proteins but also addresses the transcriptional consequences of their activity. The potential for resistance, which might arise if one pathway compensates for the loss of another, is thereby minimized. Preclinical models that explore this combination will be critical to understanding the full scope of its therapeutic potential and optimizing the strategy for clinical use.

*MYC* and *c-MYB* represent downstream targets of the KMT2A-fusion protein complex that are vital for leukemic cell survival and proliferation. The combination of menin inhibitors with strategies that suppress *MYC*, either through BET inhibitors or other indirect means, presents a multifaceted attack on the leukemic transcriptional network. MYC’s role as a central node in the oncogenic signaling network means that its inhibition could amplify the effects of menin inhibitors, leading to a collapse of the transcriptional programs that sustain leukemia. Moreover, targeting c-MYB in conjunction with menin inhibition could further disrupt the leukemic cell’s ability to maintain its malignant phenotype, particularly in the context of stem cell maintenance and differentiation blockades.

The potential for these combinations extends beyond simply additive effects; rather, they offer a synergistic approach that could lead to a complete reprogramming of leukemic cells or even their elimination. By targeting multiple points in the leukemic signaling network—ranging from chromatin modifications (DOT1L and chromatin remodeling complexes such as BRG1/BRM and KDM4C), transcriptional regulation (BRD4 and MYC) to the structural and functional disruption of the KMT2A fusion proteins themselves—these strategies aim to dismantle the leukemia at its core. Notably, the inclusion of chromatin remodeling inhibitors further enhances this approach by disrupting the chromatin landscape critical for leukemic gene expression, offering a novel avenue to complement existing therapeutic strategies. The integration of these therapies represents a new frontier in the treatment of *KMT2A*r AML, one that holds the promise of transforming the prognosis for patients who currently face limited options.

What remains now is the challenge of translating these promising preclinical findings into clinical practice. Some combinations, such as DOT1L and menin inhibitors, have already shown success and are ready for further exploration in clinical trials. Others, such as the direct targeting of KMT2A-fusion proteins or the dual inhibition of MYC and menin, are still in the early stages of investigation but offer significant potential. The critical task ahead will be to identify the most effective combinations, optimize their delivery and dosing, and carefully monitor for adverse effects. If successful, these strategies could redefine the treatment landscape for *KMT2A*r AML, offering patients a chance at durable remission and improved survival outcomes.

## 7. Conclusions

KMT2Ar leukemias continue to present a significant challenge due to their aggressive nature and high relapse rates. While menin inhibitors offer a promising therapeutic approach, the development of resistance highlights the need for more sophisticated combination strategies. This review has outlined several potential therapies, including inhibitors targeting DOT1L, BRD4, MYC, and c-MYB, as well as approaches that directly inhibit KMT2A-fusion proteins. Each of these targets addresses distinct mechanisms driving leukemogenesis, and their combination could disrupt leukemia’s reliance on single pathways, thereby overcoming resistance and improving patient outcomes.

Future efforts must focus on translating these promising preclinical combinations into clinical practice, ensuring careful optimization of delivery, dosing, and monitoring for resistance. By doing so, these approaches may significantly reshape the treatment paradigm for KMT2Ar leukemia, ultimately offering patients longer-lasting remissions and better survival outcomes.

## Figures and Tables

**Figure 1 cancers-16-04017-f001:**
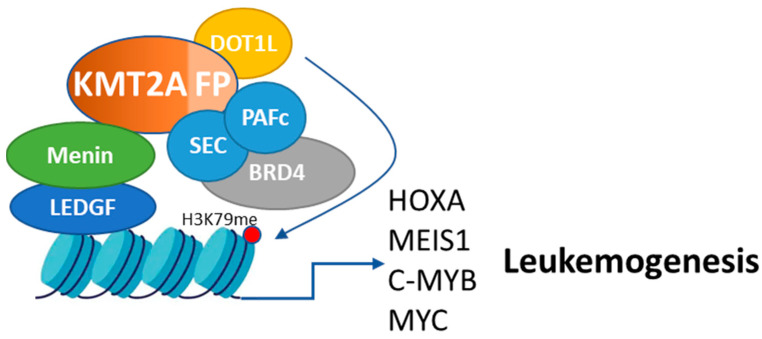
Role of KMT2A rearrangements in leukemogenesis and protein complexes of KMT2A fusion partners. The KMT2A gene, which encodes H3K4 methyltransferase, frequently undergoes chromosomal translocations, generating more than 50 distinct KMT2A fusion proteins (KMT2A-FP). These fusion proteins retain key interactions with menin, which acts as a critical scaffold by anchoring LEDGF and the PAFc complex to KMT2A fusions. This interaction facilitates the recruitment of transcriptional machinery to chromatin, driving the expression of oncogenes. Despite these retained interactions, KMT2A fusion proteins lose key regions responsible for H3K4 methylation, such as the PHD finger and SET domains, impairing normal gene regulation. Some major KMT2A fusion proteins directly recruit DOT1L, a histone methyltransferase that catalyzes H3K79 methylation. Others indirectly recruit DOT1L through p-TEFb. This misdirected DOT1L activity leads to aberrant H3K79 methylation and activation of leukemogenic genes. Additionally, most KMT2A fusion proteins recruit the super elongation complex (SEC), which, along with PAFc, interacts with BRD4, a BET family member that binds acetylated histones via its bromodomains, contributing to transcriptional dysregulation. The resulting upregulation of proto-oncogenes, including HOXA, MEIS1, c-MYB, and MYC, drives the leukemogenic process, making menin a key therapeutic target in KMT2Ar leukemias.

**Figure 2 cancers-16-04017-f002:**
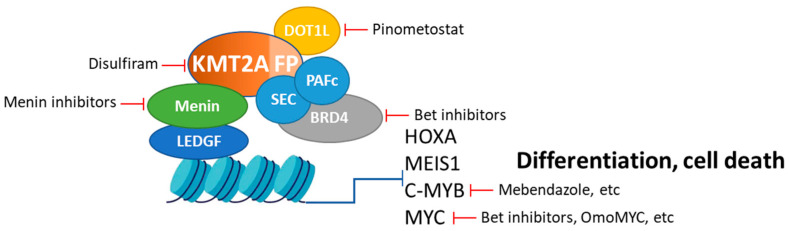
Key components and therapeutic targets in KMT2Ar leukemia. This schematic illustrates the KMT2Ar fusion complex at target gene loci. Highlighted are key components of this complex, including menin, DOT1L, and BRD4, which are essential for KMT2A fusion-mediated transformation and are targeted by small molecule inhibitors, as indicated. Also shown are the downstream targets of the KMT2A fusion protein—c-MYB, MYC—that play crucial roles in leukemogenesis. Small molecule inhibitors targeting these pathways are also indicated.

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
