# Peer review of "Synergistic Strategies for KMT2A-Rearranged Leukemias: Beyond Menin Inhibitor"

_cancers, 2024, doi:10.3390/cancers16234017_

Round 1

Reviewer 1 Report

Comments and Suggestions for Authors

The paper explores resistance mechanisms and possible new combinations for Menin inhibitors to try to improve outcomes in KMT2A-rearrenged leukemia. Despite the topic is of special interest, the review requires major revision:

-       To begin, the text repeatedly states that AMLs with NPM1 mutations have a poor prognosis, which is inaccurate. AMLs with NPM1 mutations are considered by the ELN2022 to have a favorable prognosis in the absence of FLT3 mutations, and an intermediate prognosis when they concomitantly present FLT3-ITD mutations.

-       On the other hand, the paragraph “Menin inhibitors, such as Revumenib, have shown considerable promise in treating AML characterized by KMT2Ar and NPM1mt. However, resistance to these therapies remains a significant obstacle, impacting their overall clinical effectiveness and patient outcomes. Understanding the mechanisms of resistance is crucial for devising strategies to counteract these challenges and improve therapeutic results” is repeated verbatim four times throughout the text (line 171, 177, 218 and 240), and the paragraph “DOT1L is a histone methyltransferase responsible for methylating histone H3 at lysine 79 (H3K79), a modification crucial for the transcriptional activation of genes involved in leukemogenesis[34]. In KMT2Ar leukemias, DOT1L plays a vital role in sustaining the oncogenic transcriptional programs driven by KMT2A-fusion proteins. This methylation process is essential for maintaining the expression of the downstream target genes, which promote leukemic cell survival and proliferation” appears twice (line 256 and line 263).

-       In section 4.1, which is intended to explain the genetic mechanisms of resistance to menin inhibitors, there is no explanation provided on those mechanisms.

I also have some minor comments:

-       Line: 94:  BLC-2 inhibitor venetoclax is not a mutation-specific agent, please ammeded.

-       Line 146: JNJ-75276617 now has a name; it is called Bleximenib.

-       Line 104 menin-KMT2A-LEDGF

Thank you!

Author Response

Dear Reviewer,

We sincerely thank you for your thorough review and insightful comments on our manuscript. Your feedback has been invaluable in helping us refine and improve the quality of our work. Below, we have addressed each of your comments in detail and made the necessary revisions. Line numbers and updated text are included where applicable.

Reviewer Comment 1:

To begin, the text repeatedly states that AMLs with NPM1 mutations have a poor prognosis, which is inaccurate. AMLs with NPM1 mutations are considered by the ELN2022 to have a favorable prognosis in the absence of FLT3 mutations, and an intermediate prognosis when they concomitantly present FLT3-ITD mutations.

Response:

We thank the reviewer for pointing out this inaccuracy. The text has been revised to accurately reflect the ELN2022 guidelines. The corrected text now reads:

"AMLs with NPM1 mutations are generally associated with a favorable prognosis in the absence of FLT3 mutations. However, they have an intermediate prognosis when accompanied by FLT3-ITD mutations."

This revision can be found on lines 62–65.

Reviewer Comment 2:

The paragraph "Menin inhibitors, such as Revumenib, have shown considerable promise..." is repeated verbatim four times throughout the text (lines 171, 177, 218, and 240), and the paragraph on DOT1L appears twice (lines 256 and 263).

Response:

We apologize for this oversight. The redundancy has been removed, and the text has been consolidated. The revised sections now contain the relevant information only where appropriate:

The first paragraph on menin inhibitors has been retained in sections 4 and 4.1 (lines 213–217 and 234–237).

The paragraph on DOT1L has been retained in section 5.1 (lines 300–316), and mentions in other sections have been deleted.

Reviewer Comment 3:

In section 4.1, which is intended to explain the genetic mechanisms of resistance to menin inhibitors, there is no explanation provided on those mechanisms.

Response:

We have rewritten section 4.1 to describe genetic resistance mechanisms in detail. The revised section now reads:

"Approximately 40% of cases of resistance to menin inhibitors, such as Revumenib, have been attributed to specific mutations in the MEN1 gene, particularly at residues M327, G331, or T349. These mutations decrease the binding affinity of menin inhibitors by altering critical binding sites, thereby preventing effective drug-target interaction. Structural analyses reveal that these residues are positioned near W346, a key amino acid involved in stabilizing inhibitor binding. Mutations, such as M327I, interfere with hydrogen bonds that anchor the inhibitor, causing steric clashes that reduce drug efficacy."

The updated content can be found on lines 213–237.

Minor Comments:

Comment 1: Line 94: BCL-2 inhibitor venetoclax is not a mutation-specific agent; please amend.

Response:

Thank you for highlighting this. The revised text now reads:

"Recent therapeutic developments have focused on mutation-specific agents, such as FLT3 inhibitors[15], IDH inhibitors[16], and as well as broader-targeted agents like the BCL-2 inhibitor venetoclax[17]."

This update is reflected in lines 96–97.

Comment 2: Line 146: JNJ-75276617 now has a name; it is called Bleximenib.

Response:

The name has been updated to "Bleximenib" throughout the text. This is reflected in lines 150–151.

Comment 3: Line 104: menin-KMT2A-LEDGF

Response:

The typo has been corrected. The revised text now reads "menin-KMT2A-LEDGF" in line 109.

We appreciate the time and effort you have dedicated to reviewing our manuscript. Your constructive feedback has been instrumental in strengthening our work, and we hope the revised manuscript meets your expectations. Thank you once again for your thoughtful and detailed review.

Sincerely,

Jasper de Boer

Reviewer 2 Report

Comments and Suggestions for Authors

The review article of Cantilena et al. highlights menin inhibitor development in KMT2A rearranged leukemias and NPM1 mutated AML and discusses potential therapeutic combination strategies. Overall, the paper is interesting as it treats a hot topic concerning new therapeutic developments for acute leukemias. The review is well written and comprehensive but needs some revision before publication.

1.     pp2…”NPM1-mutated AML, another subtype characterized by poor prognosis” pp3…”and NPM1 mutations (NPM1mt) continue to present significant treatment challenges, with poor prognosis and high relapse rates”:

This is not true. NPM1 mutated AML is of rather good prognosis and of intermediate prognosis if associated with FLT3-ITD mutations. This should be corrected. The introduction needs to be completed for recent ELN guidelines concerning prognosis of NPM1 mutated AML (Döhner et al. Blood 2022). Nevertheless, recurrent relapse remains an issue.

2.     pp4…”Resistance Mechanisms to Menin Inhibitors such as Revumenib, have shown considerable promise in treating AML characterized by KMT2Ar and NPM1mt. However, resistance to these therapies remains a significant obstacle, impacting their overall clinical effectiveness and patient outcomes. Understanding the mechanisms of resistance is crucial for devising strategies to counteract these challenges and improve therapeutic results.

4.1. Genetic Resistance Mechanisms

Resistance Mechanisms to Menin Inhibitors such as Revumenib, have shown considerable promise in treating AML characterized by KMT2Ar and NPM1mt. However, resistance to these therapies remains a significant obstacle, impacting their overall clinical effectiveness and patient outcomes. Understanding the mechanisms of resistance is crucial for devising strategies to counteract these challenges and improve therapeutic results.”:

This paragraph was repeated twice. Genetic resistance mechanisms should be described as they are of high interest for the reader in this paragraph.

3.     Already existing preclinical studies should be detailed and discussed:

 i.e. Zhu et al. B J Haematol 2024; Fiskus et al. Blood 2024

4.     Focus on already initiated combination trails should be discussed more in detail: reference 17 Jen et al. B J Haematol 2024

Author Response

Dear Reviewer,

We are grateful for your thoughtful review of our manuscript and for highlighting areas that required clarification and expansion. Your feedback has significantly contributed to enhancing the quality and accuracy of our work. Below, we provide a point-by-point response to your comments, with details on the revisions made and their corresponding line numbers.

Reviewer Comment 1:

The review article highlights menin inhibitor development in KMT2A-rearranged leukemias and NPM1-mutated AML. However, the prognosis for NPM1-mutated AML is misstated. The introduction needs to be completed with recent ELN guidelines concerning prognosis.

Response:

Thank you for identifying this oversight. The revised text now reads:

"AMLs with NPM1 mutations are generally associated with a favorable prognosis in the absence of FLT3 mutations. However, they have an intermediate prognosis when accompanied by FLT3-ITD mutations."

This correction, along with a reference to the ELN2022 guidelines, can be found on lines 62–65.

Reviewer Comment 2:

The paragraph on resistance mechanisms to menin inhibitors is repeated in multiple sections. Genetic resistance mechanisms should be detailed in section 4.1, as they are of high interest to the reader.

Response:

We appreciate this feedback. The redundant text has been removed, and section 4.1 has been rewritten to provide a detailed description of genetic resistance mechanisms. The revised section includes the following:

"Approximately 40% of cases of resistance to menin inhibitors, such as Revumenib, have been attributed to specific mutations in the MEN1 gene..."

The updated content is on lines 213–237.

Reviewer Comment 3:

Already existing preclinical studies (e.g., Zhu et al., Fiskus et al.) should be detailed and discussed.

Response:

We apologize for omitting these studies earlier. They have now been incorporated into a new section titled “5.6 Targeting Chromatin Remodeling” (lines 446–487), with detailed discussion on their relevance to KMT2Ar AML.

Reviewer Comment 4:

Focus on already initiated combination trials should be discussed more in detail (e.g., Jen et al., BJH 2024).

Response:

We agree and have expanded the discussion on combination trials in a new section titled “3.1 Combination Trials of Menin Inhibitors in AML” (lines 174–211). This section highlights results from trials such as KOMET-007 and SAVE, offering a comprehensive overview of this area.

Thank you for your time and effort in reviewing our manuscript. Your constructive critique has allowed us to address critical areas and improve the presentation of our work. We sincerely appreciate your valuable input and hope that the revisions meet your expectations.

Sincerely,

Jasper de Boer

Round 2

Reviewer 1 Report

Comments and Suggestions for Authors

Thank you very much for the changes made to the text. It is now better structured, and the reading flow has become easier. In order to further improve the text, I am attaching some minor changes to be made:

- Paragraph starting line 61:                 

While AMLs with NPM1 mutations are generally associated with a favorable prognosis in the absence of FLT3 mutations, their prognosis becomes intermediate when accompanied by FLT3-ITD mutations. Moreover, the prognosis of relapse and/or refractory NPM1-mutated acute myeloid leukemias is not very encouraging, with median overall survival around 5-6 months (Issa G. Blood Adv (2023) 7 (6): 933–942.) Insert new line

Menin inhibitors have shown promising preclinical activity and early clinical efficacy in patients with KMT2Ar AML and NPM1-mutated AML. However, the emergence of resistance to menin inhibitors underscores the critical need for combination therapies that target multiple oncogenic pathways to achieve more durable remissions. Insert new line

This paper explores the potential ....

- Line 96: Recent therapeutic developments have focused on mutation-specific agents, such as FLT3 inhibitors[15], IDH inhibitors[16], and as well as, broader-targeted agents such as the BCL-2 inhibitor venetoclax[17]. 

- Line 99: Remove the following sentences; they are redundant: : "NPM1-mutated AML generally carries a favorable prognosis in the absence of FLT3 mutations, although the prognosis becomes intermediate when FLT3-ITD mutations are present. Despite this, therapeutic options are limited, and the potential for relapse underscores the need for novel approaches"

- Line 213: I would avoid providing specific examples of drugs in this type of statement. “Menin inhibitors, such as Revumenib, have shown efficacy in treating”

- Line 219: I would recommend avoiding specific examples of drugs in this type of statement. Approximately 40% of cases of resistance to menin inhibitors, such as Revumenib, have been attributed to specific mutations in the MEN1 gene”

Author Response

Dear Reviewer,

Thank you for your thoughtful feedback and kind words regarding the improvements made to the manuscript after the first round of revisions. I have carefully addressed your additional comments and incorporated the suggested changes as outlined below:

Comment 1:
Paragraph starting line 61: Adjust text to include additional context regarding prognosis and menin inhibitors, and restructure the paragraph as suggested.

Response:
Thank you for this valuable suggestion. The paragraph has been revised and now reads:
*"While AMLs with NPM1 mutations are generally associated with a favorable prognosis in the absence of FLT3 mutations, their prognosis becomes intermediate when accompanied by FLT3-ITD mutations. Moreover, the prognosis of relapse and/or refractory NPM1-mutated acute myeloid leukemias is not very encouraging, with median overall survival around 5-6 months (Issa G. Blood Adv (2023) 7 (6): 933–942).

Menin inhibitors have shown promising preclinical activity and early clinical efficacy in patients with KMT2Ar AML and NPM1-mutated AML. However, the emergence of resistance to menin inhibitors underscores the critical need for combination therapies that target multiple oncogenic pathways to achieve more durable remissions."
This revision can be found on lines 60–68.

Comment 2:
Line 96: Reword the sentence for clarity: "Recent therapeutic developments have focused on mutation-specific agents, such as FLT3 inhibitors[15], IDH inhibitors[16], and as well as, broader-targeted agents such as the BCL-2 inhibitor venetoclax[17]."

Response:
The sentence has been corrected as suggested. It now reads:
"Recent therapeutic developments have focused on mutation-specific agents, such as FLT3 inhibitors[15] and IDH inhibitors[16], as well as broader-targeted agents like the BCL-2 inhibitor venetoclax[17]."

This update is on line 96.

Comment 3:
Line 99: Remove redundant sentences: "NPM1-mutated AML generally carries a favorable prognosis in the absence of FLT3 mutations, although the prognosis becomes intermediate when FLT3-ITD mutations are present. Despite this, therapeutic options are limited, and the potential for relapse underscores the need for novel approaches."

Response:
These sentences have been removed as requested.

Comment 4:
Line 213: Avoid specific examples of drugs in this type of statement: "Menin inhibitors, such as Revumenib, have shown efficacy in treating…"

Response:
The specific example has been removed, and the revised text now reads:
"Menin inhibitors have shown efficacy in treating AML characterized by KMT2Ar and NPM1 mutations."
This update is reflected in line 213.

Comment 5:
Line 219: Avoid specific examples of drugs in this type of statement: "Approximately 40% of cases of resistance to menin inhibitors, such as Revumenib, have been attributed to specific mutations in the MEN1 gene…"

Response:
The specific example has been removed. The revised sentence now reads:
"Approximately 40% of cases of resistance to menin inhibitors have been attributed to specific mutations in the MEN1 gene, particularly at residues M327, G331, or T349."
This can be found on line 219.

Thank you once again for your detailed and constructive feedback. These additional suggestions have further strengthened the manuscript, and I trust the revised version meets your expectations.

Sincerely,

Jasper de Boer
[On behalf of all authors]

Reviewer 2 Report

Comments and Suggestions for Authors

All issues were adressed.

The papier can be published in its present form.

Author Response

Dear Reviewer,

Thank you for your positive feedback and for taking the time to review our revised manuscript. We are delighted that all issues have been addressed to your satisfaction and that you find the paper suitable for publication in its present form.

Your constructive comments during the review process have been invaluable in improving the clarity and quality of our work. We greatly appreciate your thoughtful suggestions and careful evaluation.

Thank you once again for your support and contribution to the refinement of our manuscript.

Sincerely,
Jasper de Boer
[On behalf of all authors]